# Calmodulin D133H Disrupts Ca_v_1.2 and K_v_7.1 Regulation to Prolong Cardiac Action Potentials in Long QT Syndrome

**DOI:** 10.3390/cells14221763

**Published:** 2025-11-11

**Authors:** Nitika Gupta, Liam F. McCormick, Ella M. B. Richards, Kirsty Wadmore, Rachael Morris, Vanessa S. Morris, Pavel Kirilenko, Ewan D. Fowler, Caroline Dart, Nordine Helassa

**Affiliations:** 1Department of Biochemistry, Cell and Systems Biology, Institute of Systems, Molecular and Integrative Biology, Faculty of Health and Life Sciences, University of Liverpool, Liverpool L69 7ZB, UK; 2Genomic Diagnostics Laboratory, Manchester Centre for Genomic Medicine, St Mary’s Hospital, Manchester M13 9WL, UK; 3School of Biosciences, College of Biomedical and Life Sciences, Cardiff University, Cardiff CF10 3AX, UK

**Keywords:** calmodulin, D133H mutation, long QT syndrome, cardiac arrhythmia, Ca_v_1.2 (L-type calcium channel), K_v_7.1 (KCNQ1 potassium channel), calcium-dependent inactivation (CDI), ion channel regulation, calcium signalling, channelopathies

## Abstract

**Highlights:**

**What are the main findings?**

The calmodulin variant D133H disrupts Ca^2+^-dependent inactivation of Ca_v_1.2 and reduces activation of K_v_7.1.D133H reduces Ca^2+^ affinity and alters interactions with Ca_v_1.2 and K_v_7.1 binding domains.

**What are the implications of the main findings?**

The dual impact on Ca_v_1.2 and K_v_7.1 reveals cross-channel regulatory coupling as a key determinant of ventricular repolarisation.These mechanistic insights broaden understanding of how specific CaM variants remodel cardiac electrical signalling in Long QT syndrome.

**Abstract:**

Calmodulin (CaM) plays a central role in cardiac excitation–contraction coupling by regulating ion channels, including the L-type calcium (Ca^2+^) channel Ca_v_1.2 and the voltage-gated potassium (K^+^) channel K_v_7.1. Mutations in CaM are linked to severe arrhythmogenic disorders such as Long QT syndrome (LQTS), yet the molecular mechanisms remain incompletely understood. Here, we investigate the structural and functional consequences of the arrhythmia-associated CaM variant D133H. Biophysical analysis revealed that D133H destabilises Ca^2+^ binding at the C-terminal lobe of CaM, altering its Ca^2+^-dependent conformational changes. Electrophysiological recordings demonstrated that CaM D133H impairs Ca^2+^-dependent inactivation (CDI) of Ca_v_1.2, prolonging Ca^2+^ influx, while also reducing activation of K_v_7.1, thereby limiting repolarising K^+^ currents. Together, these dual defects converge to prolong action potential duration, providing a mechanistic basis for arrhythmogenesis in LQTS. Our findings establish that CaM D133H perturbs both Ca^2+^ and K^+^ channel regulation, highlighting a shared pathway by which calmodulinopathy mutations disrupt cardiac excitability.

## 1. Introduction

Long QT syndrome (LQTS) is a ventricular arrhythmic disorder with both acquired and genetic causes [1,2,3,4]. It is characterised by prolonged QT intervals on the electrocardiogram (ECG), reflecting delayed ventricular repolarisation and an extended plateau phase of the cardiac action potential (AP). These electrical disturbances predispose patients to early afterdepolarisations and torsades de pointes, which can lead to sudden cardiac death. The acquired form of LQTS is most commonly linked to pharmacological block of the hERG channel, resulting in reduced I_Kr_. In contrast, the genetic forms of LQTS, with an estimated prevalence of ~1 in 2000 [1], are associated with mutations in several ion channel genes. Over 70% of cases arise from loss-of-function mutations in the pore-forming α subunits of the voltage-gated potassium (K^+^) channels K_v_7.1 and hERG [2,5,6]. Mutations in the β subunits of these channels have also been associated with LQTS, although their pathogenic role is less definitive and may depend on additional genetic or environmental factors to manifest a clinical phenotype [7]. Gain-of-function mutations in the voltage-gated sodium channel (Na_v_1.5) [8] and the voltage-gated L-type Ca^2+^ channel (Ca_v_1.2) [9,10,11,12] are strongly linked to LQTS, causing impaired channel inactivation, enhanced depolarisation and elevated calcium (Ca^2+^) influx, which prolongs the ventricular AP plateau [11].

Recent population studies have identified multiple mutations in calmodulin (CaM), a critical Ca^2+^-binding protein, which are associated with arrhythmogenic disorders. CaM is a ubiquitous 148 amino acid Ca^2+^ sensor composed of two EF-hand-containing lobes connected by a flexible linker. It is encoded by three independent genes (*CALM1*–*3*), which produce identical proteins, although recent subcellular spatial transcript mapping suggests the genes may fulfil distinct roles [13,14]. All three are expressed in the human and animal cardiac transcriptome, underscoring the essential role of CaM in cardiac function [15,16]. In the Ca^2+^-free (apo) state, CaM adopts a “closed” conformation with high levels of disorder and flexibility [17,18]. Ca^2+^ binding induces a conformational switch to a stabilised elongated structure with exposed hydrophobic pockets in both lobes, whilst the hinge region between the helices forming the linker region ensures flexibility [19,20,21,22]. Each lobe binds two Ca^2+^ ions with different affinities (~1 μM for the C-lobe and ~10 μM for the N-lobe), enabling CaM to sense local Ca^2+^ dynamics and regulate distinct targets with both specificity and versatility [20,23,24,25,26]. CaM regulates more than 300 proteins [21], including critical cardiac ion channels. In cardiomyocytes, it mediates Ca^2+^-dependent inactivation (CDI) of Ca_v_1.2 [25], Na_v_1.5 [27,28], and the ryanodine receptor 2 (RyR2) [29], while facilitating repolarisation through the activation of K^+^ channels such as K_v_7.1 [30] and the small-conductance Ca^2+^-activated K^+^ channel, SK3 [31]. These roles make CaM indispensable for spatiotemporal control of excitation–contraction coupling and cardiac rhythm [21,32,33,34].

In the heart, Ca_v_1.2 is the major pathway for Ca^2+^ entry during the AP and plays a key role in excitation–contraction coupling (I_Ca_) [35]. Functional Ca_v_1.2 channels comprise the pore-forming α_1C_ subunit (*CACNA1C*) and the β_2_/α_2_δ_1_ (*CACNB2*/*CACNA2D1*) auxiliary subunits [36,37]. Ca_v_1.2 inactivation is controlled by both voltage-dependent inactivation (VDI), mediated by the β-subunit [37,38], and CDI, mediated by CaM [21]. CaM interaction with the IQ motif in the C-terminal domain and the NSCaTE (N-terminal Spatial Ca^2+^ Transforming Element) motif in the N-terminal domain of α_1C_ is believed to play an important role in CDI [39]. At resting intracellular Ca^2+^ concentration ([Ca^2+^]_int_), apo/CaM pre-associates with the IQ domain [40,41,42]. Upon Ca^2+^ binding, CaM interacts with both IQ [43,44,45,46,47,48,49] and NSCaTE [39,47,48,50], bridging the N- and C-termini of the channel and driving CDI. Mutations that disrupt these interactions abolish CDI, resulting in excessive Ca^2+^ entry [49,51,52,53,54].

The repolarising counterpart to I_Ca_ is the I_Ks_ current, generated by K_v_7.1 channels, which plays a key role in terminating the plateau phase. The pore-forming α subunit of K_v_7.1 (*KCNQ1*) assembles with the accessory minK subunit (*KCNE1*) [55]. CaM functions as an auxiliary subunit, required for channel assembly, trafficking, and modulation of gating [30,56]. Two CaM-binding domains (CaMBDs) on the C-terminus of the pore-forming α subunit of K_v_7.1 mediate the ion channel regulation (Helix A, HA. and Helix B, HB). Apo/CaM interacts with both domains, while Ca^2+^-bound CaM preferentially stabilises HB, promoting channel opening [57,58]. LQTS-associated mutations in HA or HB reduce current density by impairing CaM binding and destabilising K_v_7.1 regulation, suggesting a role in impaired CaM–KCNQ1 interaction in the pathophysiology of arrhythmia [59,60].

Given its essential role in regulating Ca_v_1.2, K_v_7.1, and other cardiac channels, it is unsurprising that CaM mutations cause severe arrhythmias. Since their first description in 2012, at least 25 unique arrhythmia-associated variants have been reported [16,61,62,63,64,65,66,67,68,69,70,71,72,73,74,75,76,77,78,79]. Most mutations impair Ca^2+^ binding, and can disrupt regulation of RyR2 [63,72,80,81,82,83,84,85,86], Ca_v_1.2 [15,63,65,67,69,77,84,87,88,89,90,91,92,93,94] and K_v_7.1 [70,95,96] (reviewed in [97,98,99,100,101]). Importantly, even mutations at the same position can produce distinct phenotypes, complicating predictions of functional impact across CaM variants. One example is a *CALM2* missense mutation encoding Asp133His (D133H), which alters a Ca^2+^-coordinating residue in the fourth EF-hand. This variant was identified in a 19-month-old female with LQTS who presented with foetal bradycardia, recurrent syncope, a markedly prolonged QTc (579 ms), and multiple cardiac arrests [78]. In zebrafish, the mutation causes bradycardia [96], and in vitro studies suggest impaired C-lobe binding to Ca^2+^ [78,96], mild reductions in RyR2 binding and inhibition [80], and possible alterations in K_v_7.1 trafficking [102].

However, the structural consequences of this arrhythmogenic CaM variant and its effects on Ca_v_1.2/ K_v_7.1 regulation remain poorly defined. This study investigates how the arrhythmogenic CaM variant D133H affects Ca_v_1.2 and K_v_7.1 regulation. Using structural, biochemical, and electrophysiological approaches, we show that D133H reduces C-lobe Ca^2+^ binding, disrupts CaM interaction with Ca_v_1.2 and K_v_7.1, and impairs ion channel activity. These alterations provide a mechanistic basis for prolonged ventricular APs and arrhythmia in affected patients.

## 2. Materials and Methods

**Mouse cardiac myocyte isolation.** All animal care, breeding, and experimental procedures were conducted with local ethical approval and in compliance with UK Home Office regulations and the European Parliament Directive 2010/63/EU on the protection of animals used for scientific purposes (Local Approval code BSC/21/17). Experiments were performed in adult male and female C57BL/6J mice. Mice were killed by stunning followed by cervical dislocation, and ventricular myocytes were subsequently isolated by enzymatic digestion of the heart, as previously described [103]. Briefly, the heart was removed and perfused on a Langendorff apparatus via the aorta with isolation solution containing 137 mM NaCl, 4 mM KCl, 10 mM HEPES, 10 mM creatine, 20 mM taurine, 10 mM glucose, 1 mM MgCl_2_, 50 μM CaCl_2_, 1 mg/mL collagenase Type I (Worthington Biochemical Corporation, Lakewood, NJ, USA), and 0.025 mg/mL protease XIV (Sigma Aldrich, St. Louis, MO, USA), pH 7.4, with NaOH. The entire ventricle, including left ventricle, right ventricle, and septum, was coarsely minced and then gently agitated at 35 °C. Cells were centrifuged at 50× *g* and the supernatant discarded; then, the cells were resuspended in Tyrode’s solution. Extracellular [Ca^2+^] was gradually raised in steps of 0.2, 0.5, and 1 mM.

**Molecular Biology.** For recombinant CaM protein production, human wild-type (WT) CaM was subcloned into the pE-SUMOPro-Kan vector (LifeSensors, Malvern, PA, USA) as previously reported [81,88,89,95]. Site-directed mutagenesis was performed on the CaM-WT sequence using the QuikChangeII kit (Agilent Technologies, Santa Clara, CA, USA) in accordance with the manufacturer’s protocol. The primers used to generate the D133H variant were forward 5′-GGGAAGCAGATATTGATGGTCATGGTCAAGTAAACTATGAA-3′ and reverse 5′-TTCATAGTTTACTTGACCATGACCATCAATATCTGCTTCCC-3′.

For mammalian expression of CaM in electrophysiological studies, both CaM-WT and D133H sequences were subcloned from the pE-SUMOPro-Kan vector into the pHIV-IRES-EGFP vector (Addgene, Watertown, MA, USA, plasmid #21373, gift from Bryan Welm) using NEBuilder HiFi DNA Assembly according to the manufacturer’s protocol. KCNQ1 and KCNE1 (Addgene plasmid #53048, #53050, gifts from Michael Sanguinetti) [104] were subcloned into the pHIV-IRES-dTomato vector (Addgene, plasmid #21374, a gift from Bryan Welm) as described previously [95]. In these constructs, the proteins of interest and the fluorescent reporters (EGFP or dTomato) were independently expressed under a shared promoter, allowing co-expression without forming fusion proteins. All generated constructs were verified by Sanger sequencing (MRC PPU, University of Dundee, UK).

**Electrophysiological recordings in mouse ventricular myocytes.** Purified CaM (1 µM) was dialysed into mouse ventricular myocytes via a microelectrode patch pipette for approximately 10 min, which is sufficient time and excess of CaM to replace endogenous CaM [105].

**APs**. APs were recorded in whole-cell patch clamp configuration using borosilicate glass patch pipettes (typically 1.6–2.0 MΩ) filled with an internal pipette solution containing 120 mM aspartic acid, 20 mM KCl, 10 mM K.HEPES, 10 mM NaCl, 5 mM glucose, 5 mM Mg.ATP, 10 μM EGTA, and 1 μM CaM (WT or D133H), pH 7.2. This pipette solution gave a +10 mV liquid junction potential, measured experimentally using a 3M KCl agar bridge; recordings were corrected online for this potential. APs were elicited by 2 ms current injection pulses at 1.3× threshold at 1–6 Hz. APs were recorded in 1 mM Ca^2+^ Tyrode’s solution at 32 °C (TC^2^bip, Cell MicroControls, Norfolk, VA, USA).

**I_Ca_**. Ca_v_1.2 currents were measured in mouse ventricular myocytes under a whole-cell voltage clamp. Cells were pre-treated with 1 µM ryanodine and 10 µM thapsigargin to block sarcoplasmic reticulum Ca^2+^ release [106]. Pipette solution for I_Ca_ recording contained 140 mM N-methyl-D-glucamine (NMDG), 5 mM Cs.EGTA, 10 mM HEPES, 1 mM MgCl_2_, 5 mM Mg.ATP, and 1 μM CaM (WT or D133H) (adjusted to pH 7.2, with methanesulfonic acid) [54]. Cells were patched in 1 mM Ca^2+^ Tyrode’s solution; then, the extracellular solution was exchanged using fast local perfusion (MPRE8, Cell MicroControls, VA) to I_Ca_ recording solution containing 130 mM NMDG, 1 mM MgCl_2_, 10 mM glucose, 10 mM 4-AP, 10 mM HEPES, 2 mM CaCl_2,_ or BaCl_2_, pH 7.4, with HCl at 25 °C. Cells were held at −70 mV and stepped to −50 mV for 50 ms, and then a 300 ms test pulse from −50 to +50 mV in 10 mV increments. I_Ca_ undergoes both Ca^2+^ and voltage-dependent inactivation, whereas inactivation in Ba^2+^ is mostly voltage-dependent. Thus, the fraction of inactivation due to Ca^2+^ (f50) was quantified from the proportion of peak current remaining at 50 ms with Ca^2+^ (r50_Ca_) or Ba^2+^ (r50_Ba_) as charge carrier during a step to +10 mV [107]: f50 = (r50_Ba_ − r50_Ca_)/(r50_Ba_). Cell capacitance and series resistance were compensated by 70%. Capacitive transients were subtracted using a P/4 protocol. Data were acquired at 20 kHz using a dPatch amplifier (Sutter) with Sutterpatch acquisition software.


**HEK293 cell culture, transfection, and whole-cell patch clamp electrophysiology.**


**Ca_v_1.2.** Ca_v_1.2 currents were recorded in the HEK293-Ca_v_1.2 cell line (B’SYS, Solothurn, Switzerland), transiently transfected with CaM-WT or D133H, as previously described [89]. The external (bath) solution contained 140 mM NaCl, 5 mM CsCl, 0.33 mM NaH_2_PO_4_, 5 mM glucose, 10 mM HEPES, and 1 mM MgCl_2_ (pH 7.4, adjusted with CsOH), and was supplemented with either 2 mM CaCl_2_ or BaCl_2_ for the measurement of Ca^2+^ or Ba^2+^ currents, respectively. The internal pipette solution consisted of 140 mM CsMeSO_4_, 5 mM EGTA, 10 mM Cs.HEPES, 1.91 mM CaCl_2_ (yielding 100 nM free [Ca^2+^]), 1 mM MgCl_2_, and 1 mM Na.ATP, pH 7.2. Cells were voltage-clamped at −60 mV and depolarised to test potentials ranging from −40 to +60 mV for 300 ms, with a 2 s inter-sweep interval to allow for channel recovery. CDI (f300) was quantified as the ratio of residual current at the end of the 300 ms pulse (normalised to peak current) using either Ca^2+^ or Ba^2+^ as the charge carrier (r300): f300 = (r300_Ba_ − r300_Ca_)/(r300_Ba_). Steady-state inactivation was evaluated using a 1 s conditioning pulse from −60 to +40 mV, followed by a 300 ms test pulse at +10 mV.

**K_v_7.1.** HEK293 cells were transiently transfected with CaM-IRES-EGFP, KCNQ1-IRES-dTomato, and KCNE1-IRES-dTomato, as previously described [95]. The bath solution for I_Ks_ recordings contained 140 mM NaCl, 11 mM glucose, 5.5 mM HEPES, 4 mM KCl, 1.8 mM CaCl_2_, and 1.2 mM MgCl_2_ (pH 7.4). The internal pipette solution consisted of 130 mM KCl, 10 mM HEPES, 5 mM EGTA, 1 mM Na.ATP, and 1 mM MgCl_2_, supplemented with either 100 nM or 1 μM [Ca^2+^]_free_, as calculated using Maxchelator [108]. Cells were voltage-clamped at −80 mV and depolarised to test potentials ranging from −60 to +100 mV in 20 mV increments for 4 s, followed by a 3 s repolarisation step at −40 mV.

All electrophysiology recordings from HEK293 cells were conducted at room temperature. Cells expressing CaM were identified by EGFP fluorescence. Currents were low-pass filtered at 2 kHz and digitised at 10 kHz. Patch pipettes were pulled from borosilicate glass (outer diameter of 1.5 mm and inner diameter of 1.17 mm; Harvard Apparatus, Holliston, MA, USA) and fire-polished to yield a resistance of 3–5 MΩ when filled with internal solution. Series resistance and cell capacitance were compensated. Data were analysed using Axon pClamp software (v10.7.0.3, Molecular Devices, San Jose, CA, USA) and GraphPad Prism 10.

**Expression and purification of recombinant CaM variants.** CaM variant proteins were produced recombinantly in *Escherichia coli* BL21 (DE3) STAR cells. Purification was achieved via sequential affinity and gel-filtration chromatography steps, following published protocols [81,88,89,95].

For NMR experiments, recombinant CaM proteins were labelled with ^15^N, as previously described [81,88,89,95].

**Peptides.** Peptides were designed according to the CaMBDs within Ca_v_1.2 α_1C_ (IQ domain, residues 1665–1685; NSCaTE domain, residues 51–67) [39,46] and within the C-terminal region of K_v_7.1 (HB domain, residues 507–536) [109]. Peptides were chemically synthesised and HPLC purified to >95% purity (GenicBio, Shanghai, China).Ca_v_1.2-IQ_1665−1685_: KFYATFLIQEYFRKFKKRKEQ;Ca_v_1.2-NSCaTE_51−67_: SWQAAIDAARQAKLMGS;K_v_7.1-HB_507−536_: REHHRATIKVIRRMQYFVAKKKFQQARKPY.

**Peptide/protein concentration measurements.** For binding assays, CaM protein and target peptide concentrations were quantified by measuring absorbance at 280 nm using a DS-11+ spectrophotometer (DeNovix, Wilmington, DE, USA). Molar extinction coefficients were derived from amino acid composition using the ExPASy ProtParam tool. The values used were: ε_0_ = 2980 M^−1^ cm^−1^ for CaM, Ca_v_1.2-IQ_1665−1685_, and K_v_7.1-HB_507−536_; ε_0_ = 5500 M^−1^ cm^−1^ for Ca_v_1.2-NSCaTE_51−67_.

**Equilibrium Ca^2+^ affinity titrations.** C-lobe Ca^2+^ binding affinity was assessed by monitoring intrinsic tyrosine fluorescence, as described previously [89]. CaM (5–10 μM) in 50 mM HEPES, 100 mM KCl, 1 mM MgCl_2_, 0.5 mM EGTA, and 0.5 mM NTA, pH 7.4, was titrated with CaCl_2_. Free Ca^2+^ concentrations ([Ca^2+^]_free_) were calculated using the Maxchelator program [108]. Fluorescence measurements were acquired at room temperature on a JASCO FP-6300 spectrofluorometer (λ_exc_ 277 nm, λ_em_ 300–320 nm). Calculated [Ca^2+^]_free_ were verified using the Ca^2+^-sensitive dye Cal520-FF dye (λ_exc_ 493 nm, λ_em_ 515 nm). Data were normalised and expressed as a bound fraction. The dissociation constant (*K*_d_) and Hill coefficient (*n*) were determined by fitting the data to the Hill equation using GraphPad Prism 10.

**Circular Dichroism (CD).** Far-UV CD spectra were collected at 20 °C in a 0.1 cm path length quartz cell, as previously described [95]. Spectra of CaM (10 μM) were recorded in 2 mM HEPES (pH 7.5), supplemented with either 1 mM EGTA (Ca^2+^-free, apo) or 5 mM CaCl_2_ (Ca^2+^-bound). For each condition, three scans were averaged (scan rate of 100 nm/min). After buffer subtraction, spectra were normalised to mean residual ellipticity and secondary structure content was estimated using the CDSSTR algorithm (Dichroweb, Reference set 7). The thermal stability of apo/CaM was assessed by monitoring α-helical content at 222 nm over a temperature range of 15–90 °C, with 1 °C increments, a ramp rate of 1 °C/min, and a 180 s equilibration period between steps. Normalised unfolding data were fitted to a Boltzmann sigmoid using GraphPad Prism 10 to determine the CaM melting temperature (T_m_).

**Limited Proteolysis.** The susceptibility of CaM to trypsin digestion was evaluated, as previously described [88,89,95]. Purified CaM proteins (5 μM) were incubated with trypsin for 30 min at 37 °C in 25 mM HEPES and 100 mM NaCl (pH 7.5) under either apo-conditions (10 mM EGTA, 0–10 mg/mL trypsin) or Ca^2+^-bound conditions (5 mM CaCl_2_, 0–30 mg/mL trypsin). Reactions were quenched by adding SDS sample buffer and heating at 95 °C for 10 min. Samples were separated by SDS-PAGE (NuPAGE 4–12% Bis-Tris, Life Technologies, Waltham, MA, USA) and visualised using InstantBlue staining (Abcam, Cambridge, UK). Gels were imaged on a ChemiDoc XRS+ transilluminator (Bio-Rad, Hercules, CA, USA) and the proportion of intact CaM was quantified by densitometric analysis using Fiji software [110].

**^1^H-^15^N heteronuclear single quantum coherence (HSQC) NMR spectroscopy.** NMR spectra were recorded at 303 K (30 °C) on an Avance III 800 MHz or Ascend 700 MHz spectrometer equipped with [^1^H, ^15^N]-cryoprobes (Bruker, Billerica, MA, USA). ^1^H-^15^N HSQC spectra were obtained for ^15^N-labelled CaM variants (50–100 μM) in 20 mM HEPES (pH 7.5), 50 mM NaCl, 10% (*v*/*v*) D_2_O supplemented with either 1 mM EGTA (apo/CaM) or 1 mM CaCl_2_ (Ca^2+^/CaM). Raw spectra were processed using Bruker TopSpin software, and resonance peaks were analysed and assigned using CcpNmr software [111]. Backbone assignments for CaM variants were transferred from previous work [81], and chemical shift differences were expressed in ppm as Δδ = [(ΔH)^2^ + (0.15ΔN)^2^]^1/2^.

**Isothermal Titration Calorimetry (ITC).** Experiments were carried out using CaM (50 μM) in 50 mM HEPES, 100 mM KCl, and 2 mM MgCl_2_ (pH 7.5), supplemented with either 5 mM EGTA for Ca^2+^-independent interactions or 5 mM CaCl_2_ for Ca^2+^-dependent binding interactions. Each titration comprised 20 injections of 2 μL peptide solution (prepared at a 5–10-fold molar excess relative to CaM for Ca_v_1.2 peptides and a 10–20-fold excess for K_v_7.1 peptides) into CaM samples over 4 s, with 180 s interval between injections. Measurements were performed at 25 °C with a stirring speed of 800 rpm. Heat changes were recorded on an automated PEAQ-ITC instrument (Malvern Panalytical) and analysed with MicroCal PEAQ-ITC software. Data were fitted to one-site or two-site binding models to determine the stoichiometry (N), dissociation constant (*K*_d_), enthalpy change (ΔH), and entropy change (ΔS).

**Data analysis and statistics.** All experiments were performed in a minimum of three independent replicates and data were processed using GraphPad Prism 10. Statistical comparisons were carried out using two-tailed unpaired *t*-test, one-way ANOVA, or two-way ANOVA, as indicated in the corresponding figure legends. The *p* values are denoted by asterisks as follows: * *p* < 0.05, ** *p* < 0.01, *** *p* < 0.001, and **** *p* < 0.0001. Figures were generated using CorelDRAW 2025.

## 3. Results

**D133H CaM prolongs action potential duration (APD) and promotes beat-to-beat variability in mouse ventricular myocytes.** APs were recorded in mouse ventricular myocytes dialysed with CaM-WT or D133H during continuous pacing at a 1000 ms cycle length (PCL) (Figure 1A). APD_90_ was significantly prolonged in D133H cells compared to CaM-WT cells at both 500 and 1000 ms PCL, whereas it was not different between WT and D133H cells at a shorter PCL (Figure 1B). Rate-dependent APD shortening is a normal physiological adaptation to faster heart rate; however, steeper APD_90_ restitution can precipitate a form of beat-to-beat oscillation in APD called alternans [112]. At the organ level, these can manifest as T-wave alternans, which are more common in patients with LQTS and are associated with a greater risk of life-threatening arrhythmias [113]. In the CaM-WT cell, the AP morphology is consistent, whereas in the D133H cell, the APD alternates between long and short (Figure 1C). This is more readily visualised as Poincaré plots of the APD of the current beat (APD_n_) versus the next beat (APD_n+1_) (Figure 1D). The mean absolute difference in APD of consecutive beats during 6 Hz pacing was used to quantify beat-to-beat variability. The mean beat-to-beat variability was ~2-fold greater in D133H cells (15.75 ± 5.97 ms for CaM-WT vs. 40.80 ± 9.2 ms for D133H), but this was not statistically significant (*p* = 0.08; Figure 1E).

**Acute intracellular dialysis of D133H does not affect I_Ca_ in mouse ventricular myocytes.** Purified CaM (CaM-WT or D133H) was dialysed into mouse ventricular myocytes via microelectrode patch pipette as before, and currents were elicited by step depolarisations under a voltage clamp. Following acute application of CaM variants, peak I_Ca_ recorded at +10 mV was comparable between CaM-WT (−5.33 ± 0.45 pA/pF) and D133H (−6.10 ± 0.56 pA/pF; *p* = 0.30) cells (Figure 2A). The fraction of remaining current at 50 ms (r50) in extracellular solutions containing either Ca^2+^ or Ba^2+^ revealed no significant differences (Figure 2B,C). However, there was a trend towards a reduced f50 in D133H-treated cells (0.26 ± 0.04 for CaM-WT vs. 0.18 ± 0.08 for D133H), suggesting that CDI may be affected (Figure 2C). To further characterise the effect of D133H on individual ion channels, we performed complementary experiments in a HEK293 cell model stably expressing either Ca_v_1.2 or K_v_7.1 channels.

**Voltage-dependence of Ca_v_1.2 activation and inactivation are not affected by the D133H variant.** Whole-cell patch clamp recordings were performed in HEK293 cells stably expressing Ca_v_1.2 at a resting [Ca^2+^]_int_ of 100 nM. Depolarising voltage steps were applied to generate I–V relationships (Figure 3A,B). Peak current density was slightly higher in cells expressing CaM-D133H (−3.4 ± 0.4 pA/pF) compared with CaM-WT (−2.2 ± 0.4 pA/pF), although this difference was not statistically significant (*p* = 0.0520). The voltage for half-maximal activation (V_50_) was nearly identical between groups (CaM-WT: 6.9 ± 3.8 mV and D133H: 6.9 ± 2.3 mV; *p* > 0.05) (Figure 3C). Similarly, the V_50_ of inactivation was comparable between CaM-WT (−17.0 ± 3.6 mV) and D133H (−18.0 ± 1.8 mV; *p* > 0.05) (Figure 3D,E). Collectively, these findings demonstrate that the D133H variant does not significantly impact Ca_v_1.2 voltage-dependent gating.

**CaM-mediated CDI of Ca_v_1.2 is impaired by the D133H mutation.** Ca_v_1.2 currents recorded with the D133H variant showed a pronounced increase in residual current at the end of depolarising voltage steps, consistent with reduced inactivation (Figure 3F,G). The inactivation time constants of I_Ca_ were obtained by fitting the current decay to a single exponential function. Analysis revealed that the inactivation kinetics of I_Ca_ was significantly slowed for the D133H variant, with the time constant increasing from τ = 84 ± 7 ms (CaM-WT) to τ = 448 ± 90 ms (D133H), indicating a strong impairment of Ca_v_1.2 inactivation. To distinguish between Ca^2+^-dependent and Ca^2+^-independent mechanisms, recordings were performed in the presence of Ba^2+^ as the charge carrier. Consistent with previous observations [114], the tail current kinetics for both CaM-WT and CaM-D133H did not show any significant difference between Ca^2+^ and Ba^2+^ conditions (Figure 3F,G). The proportion of persistent Ba^2+^ current (r300_Ba_) was comparable between CaM-WT (0.60 ± 0.04) and D133H (0.73 ± 0.03), indicating no difference in voltage-dependent inactivation. In contrast, the proportion of persistent Ca^2+^ current (r300_Ca_) was significantly higher with D133H (0.57 ± 0.05; *p* < 0.0001) compared with CaM-WT (0.16 ± 0.03) (Figure 3H). Correspondingly, the fraction of inactivation attributable to Ca^2+^ (f300) was reduced by ~4-fold in cells expressing CaM-D133H (0.76 ± 0.04 for CaM-WT vs. 0.21 ± 0.05 for D133H; *p* < 0.0001) (Figure 3I). These findings demonstrate that the D133H mutation severely disrupts Ca_v_1.2 inactivation by impairing CDI.

Arrhythmogenic D133H CaM variant impairs I_Ks_ at resting [**Ca^2+^**]_int_ conditions. The effect of CaM-WT and D133H on heterologously expressed K_v_7.1 currents was assessed using whole-cell patch clamp recordings under resting (100 nM) and elevated (1 µM) intracellular Ca^2+^ (Figure 4). Consistent with K_v_7.1 properties, persistent currents with minimal inactivation were observed, increasing in magnitude with stronger depolarisation (Figure 4A,D). At 100 nM [Ca^2+^]_int_, peak current density in the presence of CaM-WT was 509.6 ± 27.0 pA/pF. D133H showed significantly reduced current densities across depolarising voltages ≥+40 mV, with current at +100 mV nearly half that of CaM-WT (Figure 4B). The voltage for half-maximal activation (V_50_) was shifted from 20.1 ± 9.3 mV for CaM-WT to 42.4 ± 1.6 mV for D133H (*p* = 0.0283) (Figure 4C), indicating impaired channel activation.

At elevated [Ca^2+^]_int_ (1 µM), I–V relationships and activation kinetics were comparable between CaM-WT and D133H (Figure 4E,F). These results indicate that the D133H mutation selectively impairs K_v_7.1 current under resting Ca^2+^ conditions, whereas elevated intracellular Ca^2+^ restores normal channel function.

**Apo/CaM structure is not significantly altered by the D133H mutation.** The structural impact of the D133H mutation on apo/CaM was assessed using ^1^H-^15^N HSQC NMR (Figure 5A,B). NMR spectra were obtained for both CaM-WT and D133H, with peaks corresponding to individual residues. Due to peak heterogeneity arising from the intrinsic instability of apo/CaM, residue-level assignments were not feasible. However, spectral overlays revealed broadly similar profiles between CaM-WT and D133H, with 84% of peaks overlapping. This suggests only minor structural perturbations caused by the mutation. To further probe structural effects, CD spectroscopy was used to evaluate secondary structure content (Figure 5C,D). CaM-WT exhibited predominantly α-helical (0.38 ± 0.01) and unordered (0.36 ± 0.01) content. The secondary structure profile of D133H was comparable, with no significant differences detected when compared to CaM-WT. Collectively, these data indicate that the D133H mutation does not substantially alter the secondary structure of apo/CaM.

The D133H mutation reduces **Ca^2+^** binding affinity of CaM. Since Ca^2+^ binding is essential for CaM function, the effect of the D133H mutation was assessed by monitoring intrinsic tyrosine fluorescence from EF-hands III and IV (Figure 6A,B). D133H showed a ~ 24-fold reduced affinity for Ca^2+^ (*K*_d_ = 22 ± 2 µM) when compared to CaM-WT (*K*_d_ = 0.9 ± 0.1 µM). In addition, cooperativity of Ca^2+^ binding was reduced for D133H (*n* = 0.9 ± 0.1; *p* = 0.0002) when compared to CaM-WT (*n* = 2.5 ± 0.2). These findings demonstrate that the D133H mutation severely impairs Ca^2+^ binding at the C-lobe of CaM.

**Ca^2+^ binding** induces structural perturbations in the D133H CaM variant. Ca^2+^ binding is known to alter and stabilise the structure of CaM, which is crucial to its function. ^1^H-^15^N HSQC NMR of Ca^2+^-bound CaM-WT produced a well-dispersed spectrum with peaks of uniform intensity, consistent with a stable conformation. In contrast, D133H spectra showed heterogeneity in peak intensity, indicative of reduced stability and multiple conformational states. Spectral overlays confirmed substantial differences between the variants, with only 16.8% of residues directly overlapping and 23% assignable based on proximity to CaM-WT peaks across both lobes of the protein, indicating global structural differences (Figure 6C,D). Secondary structure analysis by CD spectroscopy revealed a decrease in α-helical content for Ca^2+^-bound D133H (0.51 ± 0.01) when compared to CaM-WT (0.59 ± 0.01; *p* < 0.0001), accompanied by a corresponding increase in unordered structures (Figure 6E,F). Together, these findings demonstrate that Ca^2+^ binding induces significant perturbations to the structure and conformational stability of D133H.

**The D133H mutation reduces CaM stability.** To assess conformational stability further, susceptibility to proteolysis by trypsin was examined under both Ca^2+^-free and Ca^2+^-saturating conditions (Figure 7A). In the absence of Ca^2+^, apo/CaM-WT and apo/D133H showed comparable digestion profiles, with near-complete degradation occurring at ~5 µg/mL trypsin (Figure 7A, left). Addition of Ca^2+^ increased resistance to proteolysis for both CaM variants, requiring ~10 µg/mL trypsin for complete digestion. However, D133H remained more susceptible, undergoing near-complete proteolysis at 5 µg/mL trypsin in the Ca^2+^-bound state (Figure 7A, right), indicating reduced conformational stability. Thermal stability of apo/CaM was assessed by CD spectroscopy at 222 nm (Figure 7B). CaM-WT exhibited a T_m_ of 44.0 ± 0.3 °C, whereas D133H showed a significantly lower T_m_ of 41.5 ± 0.3 °C (*p* = 0.0002). Together, these results demonstrate that the D133H mutation decreases the stability of CaM in both proteolytic and thermal assays.

**LQTS-associated CaM variant D133H alters binding affinities to Ca_v_1.2 CaMBDs.** ITC was used to assess the binding affinities and thermodynamic parameters of CaM-WT and D133H to Ca_v_1.2-IQ_1665−1685_ and Ca_v_1.2-NSCaTE_51−67_ domains (Figure 8). Both CaM variants bound the Ca_v_1.2-IQ_1665−1685_ domain with a 1:1 stoichiometry. However, D133H exhibited a ~4-fold higher binding affinity, with *K*_d_ decreasing from 83 ± 5 nM (CaM-WT) to 20 ± 5 nM (D133H) (Figure 8B). Although ΔG values were similar, indicating equally favourable binding, D133H displayed a more negative ΔH (−14.73 ± 0.53 kcal/mol vs. −10.54 ± 0.92 kcal/mol for CaM-WT) and a corresponding decrease in ΔS (Figure 8C). Binding affinity to Ca_v_1.2-NSCaTE_51−67_ was significantly reduced, with *K*_d_ increasing from 1.6 ± 0.3 µM for CaM-WT to 9.3 ± 1.5 µM for D133H (Figure 8D,E). This was accompanied by increased enthalpic contribution and decreased entropic contribution (Figure 8F), consistent with weaker and altered interactions. Together, these results indicate that the D133H mutation strengthens binding to the IQ domain while impairing interactions with the NSCaTE domain, reflecting domain-specific alterations in CaM-Ca_v_1.2 binding.

Disease-associated CaM variant D133H differentially affects apo- and **Ca^2+^**-dependent binding to **K_v_7.1**. ITC was used to assess the interaction of CaM-WT and D133H CaM with the C-terminal helix-binding region of K_v_7.1 (K_v_7.1-HB_507−536_). In the absence of Ca^2+^ (apo state; Figure 9A–C), CaM-WT bound K_v_7.1-HB_507−536_ with moderate affinity (*K*_d_ = 1.3 ± 0.3 µM). The reaction measured was endothermic, with an average stoichiometry of 1.0 ± 0.1. D133H displayed a >5-fold weaker interaction (*K*_d_ = 6.8 ± 1.0 µM; *p* = 0.0003), indicating reduced apo-binding affinity, along with less favourable ΔG and ΔS (Figure 9B,C). In the presence of Ca^2+^, two distinct binding events were observed at one- and two-fold molar excesses of K_v_7.1-HB_507−536_. The initial high-affinity interaction (*K*_d_ = 24 ± 3 nM) was followed by a lower-affinity interaction (*K*_d_ = 2.4 ± 0.3 µM), reflecting the Ca^2+^-dependent switch that strengthens binding (Figure 9D,E). In this Ca^2+^-bound state, D133H showed binding parameters comparable to CaM-WT for both interaction events, indicating minimal impact of the mutation under these conditions. The D133H mutation disrupts the enthalpy-driven nature of CaM’s binding to the K_v_7.1-HB_507−536_ peptide and shifts it toward a more entropy-driven interaction (Figure 9F). Overall, these data demonstrate that the D133H mutation selectively disrupts apo/CaM binding to K_v_7.1-HB_507−536_, while Ca^2+^-dependent interactions remain largely intact.

## 4. Discussion

Cardiac arrhythmias such as LQTS remain a major cause of sudden death in young individuals [115,116]. CaM variants are increasingly recognised as pathogenic drivers, yet the molecular mechanisms linking altered CaM function to arrhythmogenesis remain poorly defined. Here, we show that the D133H CaM variant impairs Ca^2+^ handling and channel regulation through reduced Ca^2+^ binding affinity, destabilisation of Ca^2+^-bound conformations, and altered interactions with Ca_v_1.2 and K_v_7.1. These defects lead to prolonged APD and increased arrhythmic risk. The key findings are summarised in Table 1, Table 2 and Table 3.

### 4.1. Structural and Biophysical Impact on CaM

NMR and CD spectroscopy confirmed that apo/D133H closely resembles CaM-WT in overall folding, consistent with other EF-hand 3−4 variants [69,70,81,82,84,88,89,93,96,117]. Nevertheless, subtle chemical shift deviations and reduced thermal stability point to local conformational changes. Separate studies exploring the thermal stability of CaM variants found no difference in apo/D133H T_m_ [96,118]. Overall, D133H did not have a major impact on apo/CaM conformation.

The ability of CaM to bind to Ca^2+^, and the resultant changes in the structure of the protein, are crucial to its ability to perform its intended functions. We observed that Ca^2+^ binding was severely impaired: affinity of the C-lobe decreased by >20-fold, from ~1 µM to ~22 µM, consistent with previous reports [78,96]. This is in line with the broader observation that EF-hand 4 variants exert disproportionately large effects on cooperativity [16,63,65,67,69,70,77,78,82,84,85,89,95]. D133H induced profound structural rearrangements upon Ca^2+^ binding, including reduced α-helical content, increased disorder, and global tertiary destabilisation. Such global effects distinguish D133H from variants such as D95V, N97I, and F141L, which exhibit localised perturbations [92,95] but resemble D131E, Q135P, and E140G [69,88,89,95]. Building on previous work showing that the D133H mutation reduces the thermal and chemical stability of Ca^2+^/CaM [96,118], we found that D133H has decreased proteolytic susceptibility. This reduction reached levels approaching that of apo/CaM, indicative of a decoupling of Ca^2+^-induced conformational change. This has also been observed in CaM variants such as D131E/H and D129G, whilst the EF3 mutants N97I and D95V have a more subtle effect [89,95,117]. Additionally, NMR heterogeneity in the C-lobe suggests D133H adopts multiple unstable Ca^2+^-bound conformations, consistent with the observed reduction in proteolytic stability. This could have major implications for CaM-mediated signalling and regulation, including interaction with K_v_7.1 and Ca_v_1.2 to modulate Ca^2+^-dependent regulation of the channel. Together, these data indicate that EF-hand 4 integrity is central not only to Ca^2+^ affinity but also to inter-lobe cooperativity and the structural transitions that enable CaM to regulate its targets.

### 4.2. Functional Consequences for Ca_v_1.2 and K_v_7.1

In Ca_v_1.2, D133H uniquely increased binding affinity for the IQ domain in high [Ca^2+^] while reducing affinity for NSCaTE. For the IQ domain, only D131E and E140G exhibited enhanced binding affinity [88,89]. In contrast, D96V, N98I, and D130G showed reduced affinity when Ca^2+^ was elevated [84,92,93,94], while F142L showed a higher affinity for IQ only under low Ca^2+^ conditions [92]. Enhanced IQ binding is rare and may explain the dominant inheritance of D133H-associated LQTS, where disease phenotype is observed despite only one CaM-encoding allele out of six being affected: the mutant could outcompete CaM-WT at this critical regulatory site. In addition to D133H, the Q135P and E140G variants, but not D131E, showed reduced affinity for NSCaTE, indicating that even mutations located within the same EF-hand can have divergent effects on channel interaction [88,89]. Physiologically, LQTS-associated CaM mutations that have been probed for effects on Ca_v_1.2 regulation have demonstrated a reduction in CDI to varying extents [15,65,67,69,70,77,87,90,91,119]. Here, we have shown that D133H impaired Ca_v_1.2 CDI, consistent with other LQTS variants [67,69,88,89,90] but in contrast to CPVT-associated CaM mutations, which generally preserve CDI [61,63,90]. Together with the observed APD prolongation, these results support a shared pathogenic mechanism across LQTS-associated CaM variants. Interestingly, previous computational studies have demonstrated that the LQTS-associated G406R mutation in the Ca_v_1.2 α-subunit can also prolong the cardiac action potential and increase arrhythmogenic risk by impairing inactivation of the L-type Ca^2+^ current (I_Ca_) [120]. Our findings extend this concept by showing that a mutation in calmodulin (D133H), a regulatory protein rather than the channel pore-forming subunit, can produce a similar electrophysiological phenotype through a distinct molecular mechanism.

We have shown that D133H reduces binding to the K_v_7.1 HB domain in the absence of Ca^2+^, but not under saturating Ca^2+^, mirroring functional reductions in I_Ks_ density and a depolarising shift in activation under resting [Ca^2+^]_int_. This pattern differs from D95V and D131H, which impair binding and regulation under both apo and elevated Ca^2+^ conditions [95]. Conversely, a separate study found no change in either the rate or voltage-dependence of the activation of I_Ks_ modulated by D133H in an oocyte model [102]; however, the intracellular Ca^2+^ concentration there was not determined. Our findings indicate a more nuanced regulation of I_Ks_ dependent on Ca^2+^. Prior FRET-based assays reported enhanced overall binding of D133H to the intact channel [102], likely reflecting methodological differences: modestly altered interactions at HB may be obscured when total channel–CaM interaction is measured. Notably, the use of cerulean-tagged CaM in that FRET assay adds a ~27 kDa fluorophore, which could alter the structure–function of CaM, thereby influencing the observed binding behaviour. Our results suggest that, while D133H does not abolish binding, it perturbs the specific apo-state interaction needed to prime K_v_7.1 for rapid activation, thereby delaying I_Ks_ onset. Because murine myocytes are a poor model for I_Ks_ [121], human iPSC-derived cardiomyocytes or larger animal models will be essential to clarify these effects in vivo.

### 4.3. Electrophysiological Mechanisms

D133H prolonged APD_90_ in ventricular myocytes, most notably at longer pacing cycle lengths. The resultant APD lengthening can be explained by two coupled mechanisms: (i) increased I_Ca_ during the plateau opposing repolarising currents, and (ii) enhanced Ca^2+^ entry and SR loading, which increase Ca^2+^ transient amplitude/duration and inward I_NCX_ (Na^+^-Ca^2+^ exchanger) [122]. Given that I_Ca_ at the murine plateau (~−40 mV) is minimal, whereas I_NCX_ is prominent during incomplete cytosolic Ca^2+^ recovery [103], we favour the latter mechanism as the primary driver under our experimental conditions. This interpretation contrasts with dystrophin-deficient (Mdx) mice, in which reduced I_Ca_ CDI did not prolong APD due to strong Ca^2+^ buffering with EGTA, despite mice exhibiting prolonged QTc intervals in vivo [123]. Our conditions preserved physiological Ca^2+^-V_m_ (membrane potential) coupling by minimising exogenous buffering, revealing the contribution of I_NCX_. In humans, ventricular myocytes exhibit more positive plateau potentials that sustain I_Ca_ and directly prolong APD [124], while simultaneously limiting I_NCX_. Adult murine ventricular myocytes possess little or no I_Ks_, and thus the impact of D133H on this current would not contribute to the APD lengthening we observed [121], whereas this is an important repolarising current in humans. Thus, an equivalent reduction in CDI or inhibition of I_Ks_ is likely to cause more pronounced APD prolongation in human versus murine cells [123]. While the precise mechanism underlying APD lengthening may vary depending on species, reduced CDI resulting from D133H confers a pro-arrhythmic phenotype. Increased T-wave alternans are associated with greater risk of life-threatening arrhythmias in LQTS patients [113]. At the tissue level, spatially discordant electrical alternans can cause unidirectional conduction block and the initiation of re-entrant arrhythmias. D133H caused APD lengthening at longer PCL but not at the shortest PCL, resulting in steeper APD restitution. However, we did not find a significant difference in beat-to-beat APD_90_ variability between CaM-WT and D133H cells during 6 Hz pacing. This indicates that arrhythmogenicity may stem more from restitution dynamics and APD prolongation than from short-term variability.

### 4.4. Broader Implications

The net arrhythmogenic phenotype of D133H likely reflects impaired I_Ks_ activation combined with reduced Ca_v_1.2 CDI during sustained Ca^2+^ entry. Calmodulin is a ubiquitous signalling molecule capable of interacting with numerous intracellular targets, including the plasma membrane Ca^2+^-ATPase (PMCA), Na^+^/Ca^2+^ exchanger (NCX), Ca^2+^/CaM-dependent protein kinase II (CaMKII), ryanodine receptor 2 (RyR2), Na_v_1.5 and gap junction proteins [125,126,127,128,129,130,131,132]. Our cellular and peptide-binding data indicate that the functional alterations observed with the D133H include direct modulation of Ca_v_1.2 and K_v_7.1. Secondary effects on other cardiac targets could also occur, and may either amplify or attenuate the pro-arrhythmic consequences of the D133H mutation. Previous studies have reported modest reductions in RyR2 inhibition [80,133] but no changes in SK3 channel regulation [31], while the broad expression of CaM suggests that extra-cardiac manifestations may also occur, as evidenced by distinct neuronal phenotypes of arrhythmia-associated CaM variants in *C. elegans* [134]. Collectively, these findings emphasize the multifaceted influence of CaM on cardiac excitability and highlight the therapeutic potential of strategies aimed at stabilising specific CaM–channel interactions.

Overall, this work contributes to the growing body of evidence exploring the unique and common pathophysiological effects of different CaM variant-mediated arrhythmias, aiding the development of more effective treatments for patients with these disorders. Emerging gene therapy approaches, including allele-specific knockdown or CRISPR editing, have shown promise [91,135], but strategies such as peptide competitors which selectively disrupt mutant CaM binding to Ca_v_1.2, may represent a more tractable alternative [136].

## 5. Conclusions

To summarise, this study provides nuanced mechanistic insights into the possible factors leading to CaM-mediated LQTS. D133H CaM destabilises Ca^2+^-bound conformations, reduces Ca^2+^ affinity, and alters binding to Ca_v_1.2 and K_v_7.1 in a manner that prolongs APD and promotes arrhythmia. Enhanced binding of D133H to Ca_v_1.2-IQ, impaired binding to Ca_v_1.2-NSCaTE, and reduced apo-state interaction with K_v_7.1 together provide a mechanistic explanation for the dominant arrhythmogenic phenotype (reduced I_Ks_ and impaired I_Ca_ CDI). These data reinforce the critical role of EF-hand 4 in CaM function and extend our understanding of how specific structural perturbations propagate to cellular and tissue-level electrophysiology in LQTS.

## Figures and Tables

**Figure 1 cells-14-01763-f001:**
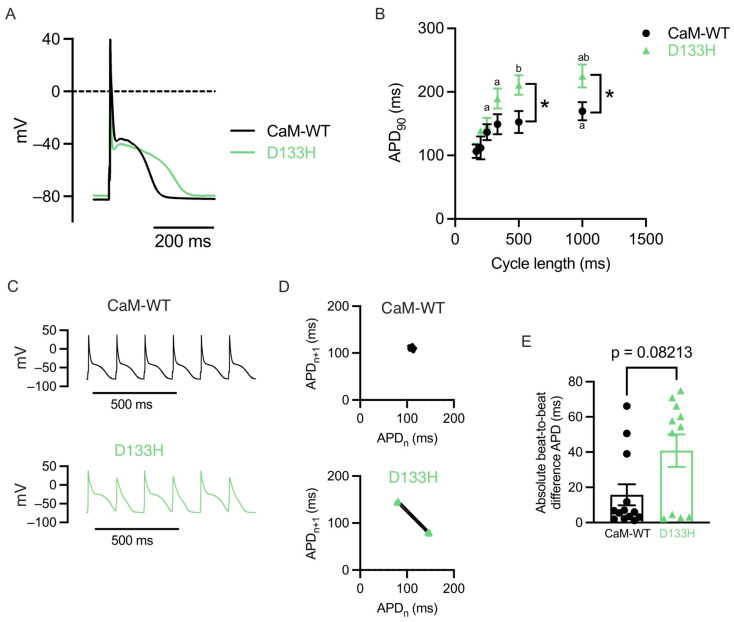
LQTS-associated CaM variant D133H prolongs APD in mouse ventricular myocytes. (**A**) Exemplar APs recorded at 1 Hz in cells dialysed with either CaM-WT (black line) or D133H (green line). (**B**) APD_90_ in CaM-WT and D133H cells during steady pacing at different cycle lengths. APD_90_ was longer in D133H cells at cycle lengths of 500 and 1000 ms. * *p* < 0.05 CaM-WT vs. D133H, ^a^
*p* < 0.05 vs. 166 ms, ^b^
*p* < 0.05 vs. 250 ms. Mixed-effects model with Šídák’s multiple comparisons test. (**C**) Exemplar APs recorded during pacing at 166 ms cycle length in a CaM-WT and D133H cells. (**D**) Poincaré plots showing the relationship between APD (APD_n_), and the APD of the next beat (APD_n+1_) constructed from the data shown in panel C. Two clusters of APD values occur in the D133H cell, indicating beat-to-beat variation in repolarisation. (**E**) The absolute difference in APD between consecutive beats during pacing at 166 ms cycle length was around 2× greater in D133H cells; however, this was not statistically significant (*p* = 0.08; Mann–Whitney test). (B + E) n/N = 13/4 CaM-WT cells/hearts and n/N = 11/4 D133H cells/hearts.

**Figure 2 cells-14-01763-f002:**
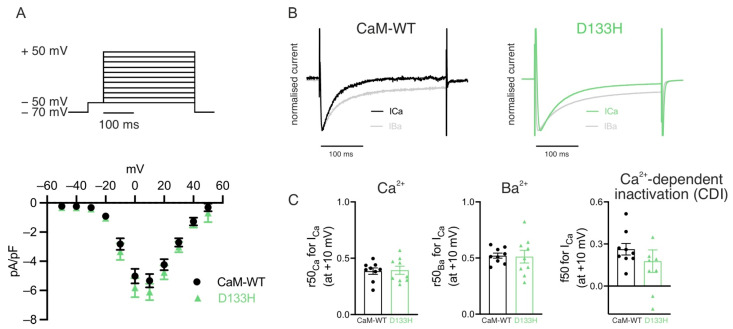
**Effects of arrhythmogenic CaM variant D133H on I_Ca_ in mouse ventricular myocytes.** (**A**) I_Ca_ current–voltage (I–V) relationship in patch-clamped ventricular myocytes dialysed with either 1 µM CaM-WT (white symbols) or D133H (green symbols). Top panel illustrates the voltage clamp protocol used. Peak I_Ca_ at +10 mV was not different between CaM-WT and D133H cells (*p* = 0.30; unpaired *t*-test). (**B**) Exemplar normalised current recordings in a CaM-WT (left) and D133H (right) cells during a 300 ms step to +10 mV, with 2 mM Ca^2+^ or 2 mM Ba^2+^ as the divalent cation. (**C**) Quantification of CDI from the proportion of remaining current at 50 ms in Ca^2+^ and Ba^2+^. n/N = 9/4 CaM-WT cells/hearts and n/N = 9/4 D133H cells/hearts.

**Figure 3 cells-14-01763-f003:**
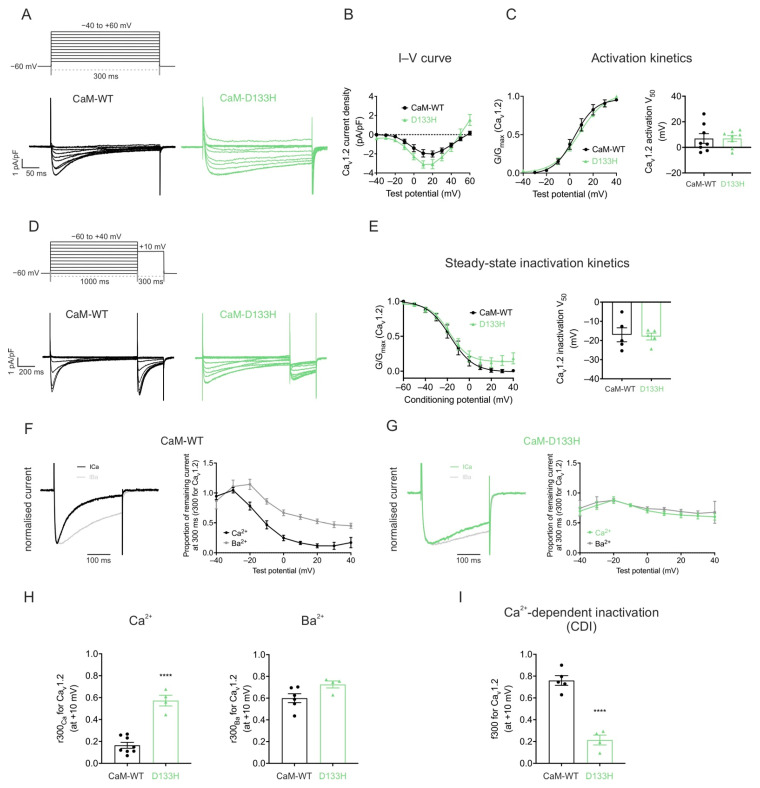
**Arrhythmogenic CaM variant D133H disrupts CDI of Ca_v_1.2 in HEK293 cells.** (**A**) Voltage step protocol for Ca_v_1.2 activation. Cells were depolarised from −40 to +60 mV in 10 mV increments for 300 ms from a holding potential of −60 mV. Representative traces from HEK293-Ca_v_1.2 cells expressing CaM variants are shown. (**B**) Current–voltage (I–V) relationships and (**C**) activation curves for Ca_v_1.2 with CaM variants. Data were normalised to the peak current density for each cell. Conductance, G, was normalised to peak conductance, G_max_, to give average activation curves. Half-maximal activation voltages, V_50_, were determined from Boltzmann fits (CaM-WT, n  =  8; D133H, n  =  8). (**D**) Voltage step protocol for inactivation. Cells were subjected to 1000 ms conditioning pulses from −60 to +40 mV in 10 mV increments, followed by a 300 ms test pulse to +10 mV. Representative traces from CaM-variant expressing cells are shown. (**E**) Steady-state inactivation curves of Ca_v_1.2 (CaM-WT, n  =  5; D133H, n  =  5). (**F**,**G**) (Left panels) Representative Ca^2+^ and Ba^2+^ current traces, normalised to their respective peak current, in response to a 300 ms pulse to +10 mV. (Right panels) Fractional residual Ca^2+^ and Ba^2+^ current (r300), at test potentials from −40 to +40 mV. (CaM-WT Ca^2+^, n  =  8; Ba^2+^, n  =  6; D133H Ca^2+^, n  =  13; Ba^2+^, n = 4). (**H**) Residual currents at +10 mV (r300_Ca_ and r300_Ba_) (CaM-WT Ca^2+^, n  =  8; Ba^2+^, n  =  6; D133H Ca^2+^, n  =  4; Ba^2+^, n  =  4). (**I**) Proportion of inactivation due to CDI (f300), calculated as (r300_Ba_ − r300_Ca_)/r300_Ba_ at +10 mV (CaM-WT, n  =  5; D133H, n  =  4). Data are mean ± s.e.m. and statistical differences were determined using a two-tailed unpaired *t*-test (**** *p* < 0.0001).

**Figure 4 cells-14-01763-f004:**
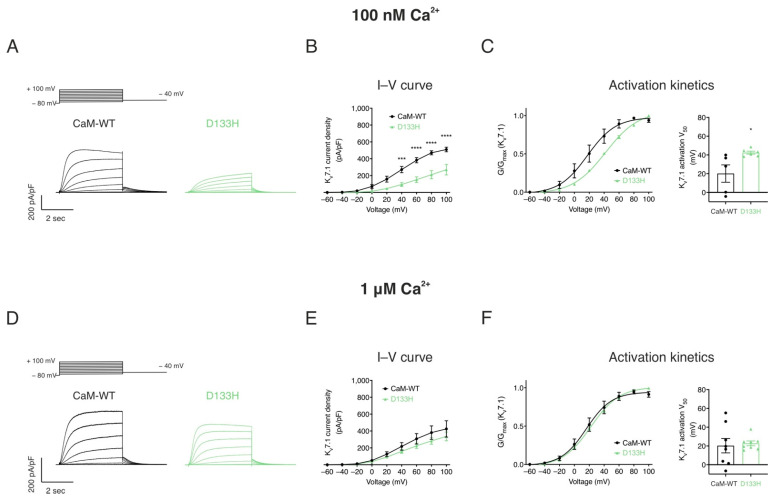
**LQTS-associated CaM variant D133H reduces I_Ks_ densities and shifts voltage sensitivity at resting intracellular [Ca^2+^].** HEK293 cells were transiently co-transfected with KCNQ1, KCNE1, and CaM variants. Whole-cell voltage clamp recordings were performed by holding cells at −80 mV and applying 4 s depolarising steps from −60 to +100 mV in 20 mV increments, followed by repolarisation to −40 mV. Currents were recorded at two intracellular Ca^2+^ concentrations: 100 nM and 1 μM. (**A**,**D**) Representative I_Ks_ current traces at (**A**) 100 nM and (**D**) 1 µM [Ca^2+^]_int_. (**B**,**E**) Current–voltage (I–V) relationships of I_Ks_ at (**B**) 100 nM and (**E**) 1 μM [Ca^2+^]_int_. (**C**,**F**) Activation kinetics at (**C**) 100 nM and (F) 1 μM [Ca^2+^]_int_. Left panels: Data were normalised to the peak current density for each cell. Conductance, G, was normalised to peak conductance, G_max_, to give average activation curves. Right panels: half-maximal activation voltage (V_50_) derived from Boltzmann fits. Data are mean  ±  s.e.m. (CaM-WT, n  =  5; D133H, n  =  6). Differences between groups were determined using two-way ANOVA (I–V) or one-way ANOVA (activation kinetics) with Dunnett’s multiple comparisons test (* *p*  <  0.05, *** *p*  <  0.001, and **** *p*  <  0.0001).

**Figure 5 cells-14-01763-f005:**
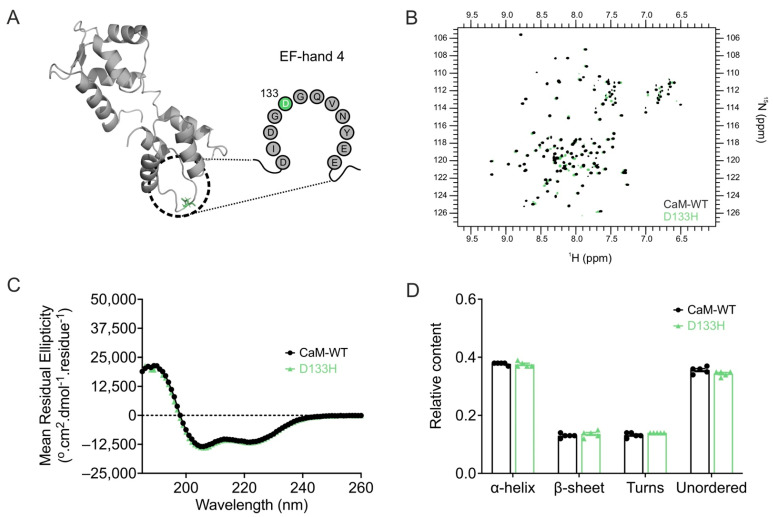
**Arrhythmia-associated mutation D133H does not alter apo/CaM structure.** (**A**) Cartoon representation of apo/CaM (PDB 1CFC), highlighting the D133H mutation (green) within the EF-hand motif. (**B**) Overlay of ^1^H-^15^N HSQC NMR spectra for apo/CaM-WT and D133H variants. Each cross-peak represents an amide proton–nitrogen pair, with positions defined by chemical shifts along the ^1^H (x-axis) and ^15^N (y-axis) dimensions. (**C**) Far-UV CD spectra and (**D**) estimated secondary structure content from the CDSSTR algorithm (reference data set 7). Data are mean  ±  s.e.m. (CaM-WT, *n* = 5; D133H, *n* = 5), and statistical differences were assessed by two-way ANOVA followed by Dunnett’s multiple comparisons test.

**Figure 6 cells-14-01763-f006:**
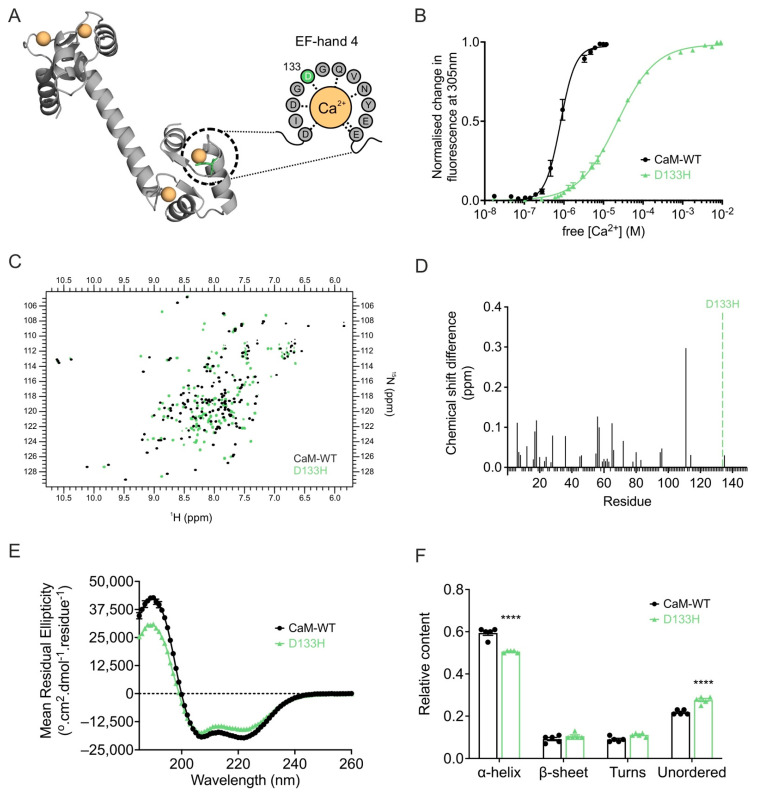
**LQTS-associated CaM variant D133H exhibits decreased Ca^2+^ binding and altered Ca^2+^/CaM structure.** (**A**) Cartoon representation of Ca^2+^/CaM (PDB 1CLL) highlighting the D133H mutation (green) in the Ca^2+^-coordinating EF-hand. Dashed grey lines indicate interactions between residues and Ca^2+^. (**B**) C-lobe Ca^2+^ affinity of CaM measured by intrinsic tyrosine fluorescence. Data were fitted using a specific binding model with Hill slope to calculate *K*_d_ values (CaM-WT, n = 7; D133H, n = 5). (**C**) Overlay of ^1^H-^15^N HSQC NMR spectra of Ca^2+^/CaM-WT and D133H variant. Each cross-peak corresponds to an amide proton–nitrogen pair positioned by its chemical shift in both ^1^H and ^15^N dimensions. (**D**) Chemical shift analysis of Ca^2+^/CaM-WT and D133H variant. Histograms show chemical shift difference for each residue; unassigned residues were given a uniform value of −0.01. The position of the mutation is indicated by a green dotted line. (**E**) Far-UV CD spectra for Ca^2+^/CaM variants and (**F**) secondary structure estimations using the CDSSTR algorithm (reference data set 7). Data are mean ± s.e.m. (CaM-WT, n = 5; D133H, n = 5), and differences between groups were determined using a two-tailed unpaired *t*-test (**** *p* < 0.0001).

**Figure 7 cells-14-01763-f007:**
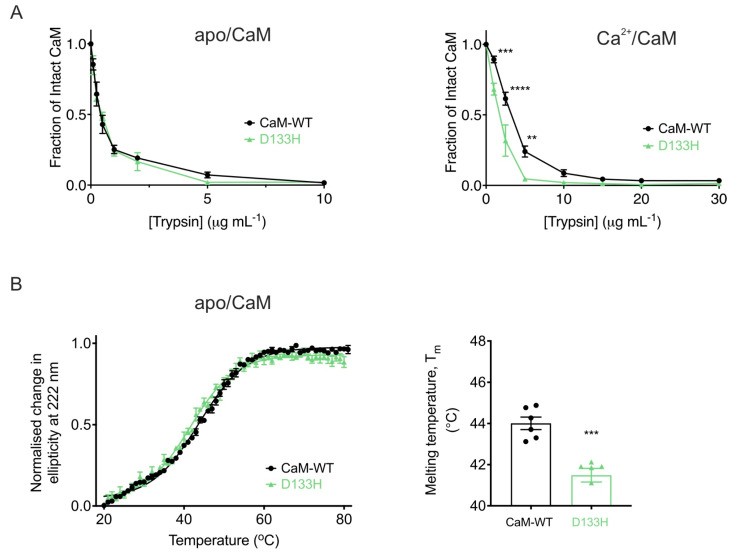
**LQTS-associated mutation D133H decreases CaM stability.** (**A**) Limited proteolysis of CaM variants in (left) EGTA (10 mM) or (right) CaCl_2_ (5 mM). Purified CaM proteins were incubated with increasing trypsin concentrations for 30 min at 37 °C. Remaining intact CaM was quantified via SDS-PAGE and densitometry (Fiji), normalised to control. Data are mean ± s.e.m. (apo/CaM-WT, n  =  6; apo/D133H, n  =  7; Ca^2+^/CaM-WT, n  =  7; Ca^2+^/D133H, n  =  6). Statistical analysis was performed using two-way ANOVA with Dunnett’s multiple comparisons test. (**B**) Thermal denaturation of apo/CaM proteins monitored by circular dichroism spectroscopy (λ = 222 nm). (left) Average unfolding curves were fitted to the Boltzmann sigmoid equation. (right) Melting temperatures (T_m_) were derived from half-maximal unfolding. Data are mean ± s.e.m (CaM-WT, n  =  6; D133H, n  =  6). Statistical difference between groups was assessed with a two-tailed unpaired *t*-test (** *p*  <  0.01, *** *p*  <  0.001, and **** *p*  <  0.0001).

**Figure 8 cells-14-01763-f008:**
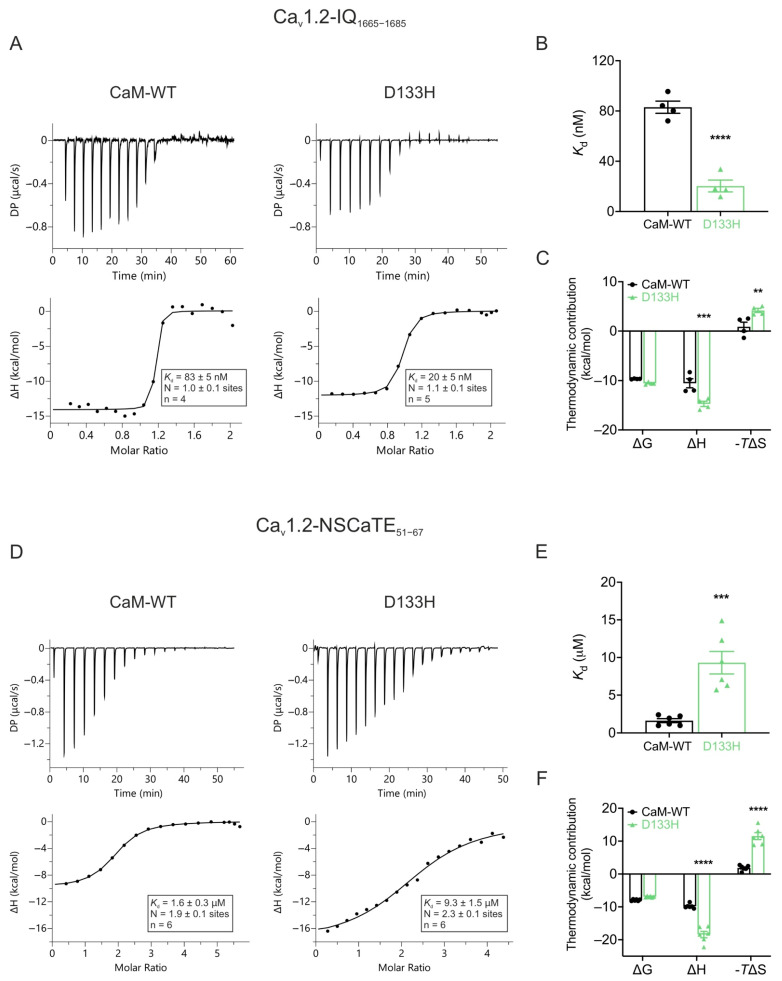
**Arrhythmia-associated CaM variant D133H shows altered binding to Ca_v_1.2 IQ and NSCaTE domains.** (**A**,**D**) Example ITC titration profiles illustrating raw heat signals (upper panels) and integrated isotherms after baseline correction (lower panels) for interactions with (**A**) Ca_v_1.2-IQ_1665−1685_ and (**D**) Ca_v_1.2-NSCaTE_51−67_. (**B**,**E**) Binding affinity of Ca^2+^/CaM-WT and D133H to Ca_v_1.2-IQ_1665−1685_ and Ca_v_1.2-NSCaTE_51−67_, respectively. (**C**,**F**) Thermodynamic profile of these interactions, including Gibbs free energy change (ΔG), enthalpy change (ΔH), and entropy contribution (-TΔS). DP, differential power. Data are mean  ±  s.e.m. (IQ: CaM-WT, n = 4, D133H, n = 5; NSCaTE: CaM-WT, n = 6, D133H, n = 6). Differences between groups were determined using a two-tailed unpaired *t*-test (affinity) and two-way ANOVA with Dunnett’s multiple comparisons test (thermodynamics) (** *p*  <  0.01, *** *p*  <  0.001, and **** *p*  <  0.0001).

**Figure 9 cells-14-01763-f009:**
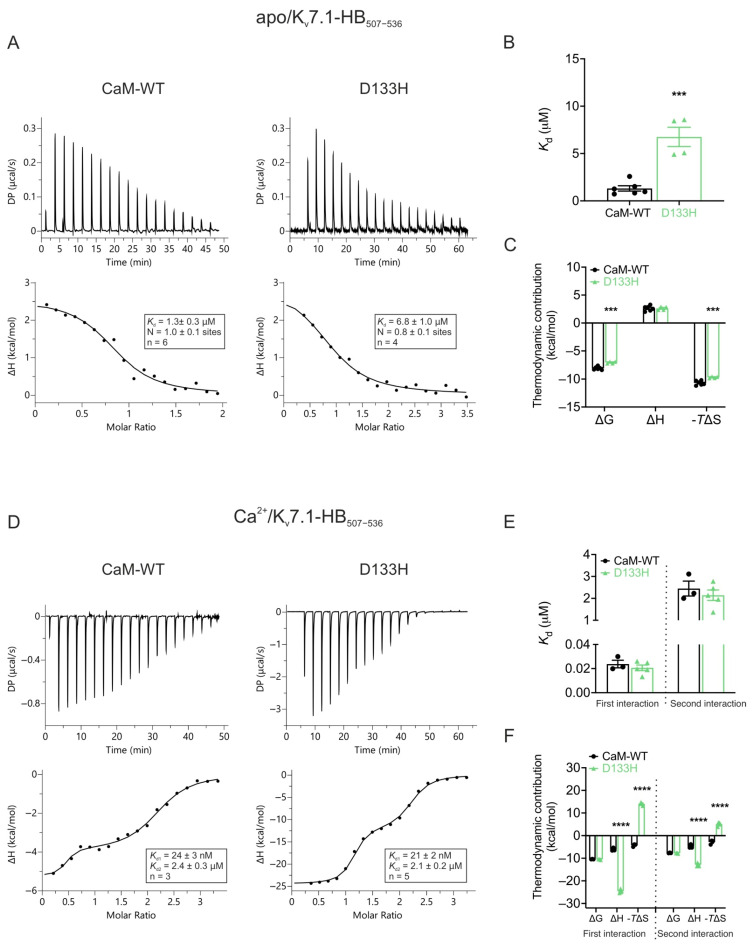
**Apo/CaM binding to K_v_7.1-HB_507_**_−**536**_**is decreased for LQTS-associated variant D133H.** (**A**) Example ITC titration profiles illustrating raw heat signals (upper panel) and the corresponding binding isotherms after baseline correction (lower panels) for the interaction of apo/CaM with K_v_7.1-HB_507−536_. (**B**) Binding affinity and (**C**) thermodynamic parameters for apo/CaM interaction with K_v_7.1-HB_507−536_ obtained by fitting the data to a single-site binding model. (**D**) Representative ITC titration curves (upper panel) and binding isotherms (lower panel) for the interaction between Ca^2+^/CaM and K_v_7.1-HB_507−536_. (**E**) Affinity and (**F**) thermodynamic profile of the binding of Ca^2+^/CaM to K_v_7.1-HB_507−536_ obtained by fitting to a two-site binding model. Data are means ± s.e.m. N, stoichiometry; n, number of experimental replicates. The change in free energy (ΔG) was calculated as ΔH − TΔS, where ΔH represents enthalpy change, ΔS is entropy change, and T is absolute temperature. DP, differential power. Statistical analyses were performed using a two-tailed unpaired *t*-test for affinity data and two-way ANOVA with Dunnett’s multiple comparisons for thermodynamic parameters (*** *p*  <  0.001 and **** *p*  <  0.0001).

**Table 1 cells-14-01763-t001:** Summary of functional effects of the LQTS-associated CaM variant D133H on Ca_v_1.2 activity. Values are mean ± s.e.m.

	Ca_v_1.2 (Patch Clamp Electrophysiology)						
	Stable HEK293					Mouse Cardiomyocyte	
	Peak Current Density (pA/pF)	Reversal Potential (mV)	V_50_ Activation (mV)	V_50_ Inactivation (mV)	CDI (f300)	APD_90_ at 1 Hz (ms)	CDI (f50)
CaM-WT	−2.2 ± 0.4	58.7 ± 2.8	6.9 ± 3.8	−17.0 ± 3.6	0.76 ± 0.04	170 ± 14	0.26 ± 0.04
D133H	−3.4 ± 0.4	52.4 ± 1.6	6.9 ± 2.3	−18.0 ± 1.8	0.21 ± 0.05 ****	225 ± 18 *	0.18 ± 0.08

* *p*  <  0.05 and **** *p*  <  0.0001 vs. CaM-WT.

**Table 2 cells-14-01763-t002:** Summary of functional effects of the LQTS-associated CaM variant D133H on K_v_7.1 activity. Values are mean ± s.e.m.

	K_v_7.1 (Patch Clamp Electrophysiology)			
	100 nM Ca^2+^		1 μM Ca^2+^	
	Current Density at +100 mV (pA/pF)	V_50_ Activation (mV)	Current Density at +100 mV (pA/pF)	V_50_ Activation (mV)
CaM-WT	509.6 ± 27.0	20.1 ± 9.3	424.7 ± 97.0	20.3 ± 7.7
D133H	270.2 ± 62.2 ****	42.4 ± 1.6 *	321.2 ± 47.3	22.8 ± 2.5

* *p*  <  0.05 and **** *p*  <  0.0001 vs. CaM-WT.

**Table 3 cells-14-01763-t003:** Summary of binding affinities of LQTS-associated CaM variant D133H for Ca^2+^ and ion channel CaMBDs. *K*_d_ values (μM), mean ± s.e.m.

	Ca^2+^	Ca_v_1.2		K_v_7.1	
		IQ_1665−1685_	NSCaTE_51−67_	HB_507−536_ (apo)	HB_507−536_ (Ca^2+^)
CaM-WT	0.9 ± 0.1	0.083 ± 0.005	1.6 ± 0.3	1.3 ± 0.3	0.024 ± 0.003 2.4 ± 0.3
D133H	22.2 ± 1.5 ****	0.020 ± 0.005 ****	9.3 ± 1.5 ***	6.8 ± 1.0 ***	0.021 ± 0.002 2.1 ± 0.2

*** *p*  <  0.001 and **** *p*  <  0.0001 vs. CaM-WT.

## Data Availability

The original contributions presented in this study are included in the article. Further inquiries can be directed to the corresponding author.

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
