# Peer review of "Calmodulin D133H Disrupts Cav1.2 and Kv7.1 Regulation to Prolong Cardiac Action Potentials in Long QT Syndrome"

_cells, 2025, doi:10.3390/cells14221763_

Round 1
Reviewer 1 Report
Comments and Suggestions for Authors
The present study by Drs. Gupta, Maccormick and colleagues describes functional consequences of a calmodulin (CaM) mutation, D133H, on CaV1.2 and KV7.1 channel activities. The study aims to reveal a molecular mechanism of how CaM-D133H may contribute to cardiac repolarization disorders. The paper was well written, and the experiments have been thoroughly designed.
Calmodulin (CaM) is a modulator of channel gating and previous publications had shown that it mediates Ca2+-dependent inactivation (CDI) of CaV1.2 channels and Ca2+-dependent facilitation (CDF) of KV7.1 channels. Here the authors show that CaM-D133H prolongs the action potential duration (APD) in mouse ventricular myocytes, which is a substrate for long QT syndrome. The authors further show that ventricular ICa did not display CDI in the presence of the CaM mutant, which suggests that an increase in Ca2+ influx could cause the observed APD prolongation. A separate set of electrophysiological experiments in HEK 293 cells confirmed that CaM-D133H increased the amplitude of ICa, attenuated its CDI and removed CDF of IKs. Mechanistically the authors provide data which show that the mutation reduced the affinity of CaM for Ca2+ ions, which altered its active conformation and the binding modes of CaM to calmodulin binding domains of both ion channels.
Whereas the effects of CaM-D133C on recombinant ICa were well pronounced in HEK 293 cells where the slower inactivation kinetics prolonged Ca2+ influx (remaining ICa at r300 in Figure 3H), the functional consequences of the CaM mutant on endogenous ICa and its contribution to APD prolongation in ventricular myocytes were less well defined. Because there was no relevant increase in peak Ca2+ current (Figure 2A), the proposed increase in Ca2+ influx can only arise from slower inactivation of the current. However, the key analysis, a comparison of the inactivation kinetics of ICa that shows that the current inactivates slower in the presence of CAM-D133H as compared to wild type CaM is missing in Figure 2. Please provide an analysis of current inactivation properties as shown in Figures 3F through 3I or provide tau-values for the inactivation kinetics of ICa for both experimental conditions in ventricular myocytes.
Reviewer 2 Report
Comments and Suggestions for Authors
In this study, the investigators explored the mechanism of arrhythmogenesis caused by the Calmodulin (CaM) variant D133H, which is linked to Long QT Syndrome (LQTS). Biophysical analysis showed that the D133H mutation might destabilize Ca2+ binding at CaM's C-terminal lobe, thereby altering its normal Ca2+-dependent shape change. Functionally, this variant could cause dual defects in cardiac ion channel regulation: it impairs the Ca2+-dependent inactivation (CDI) of the L-type Ca2+ channel (Cav1.2), leading to prolonged Ca2+ influx, and it reduces the activation of the repolarizing K+ channel (Kv7.1). These combined effects were consequently thought to prolong the cardiac action potential duration, establishing a clear molecular mechanism for the observed arrhythmias and highlighting how CaM mutations disrupt cardiac excitability by affecting both Ca2+ and K+ channel regulation. The results in this study appear to be interesting. Several inquiries regarding this manuscript have been shown as follows.
Major comments:
- When calmodulin or its mutants are included in the pipette solution for whole-cell patch-clamp current or potential recordings performed in this study, these peptides or proteins may also interact with and influence other intracellular components, subsequently affecting the function or kinetics of CaV1 or KV7.1 channels.
- In Figures 2B, please show the trail current as the potential was returned to -70 mV. In Figure 3G, the current trace for Ba2+ current could be incorrect, particularly at the tail deactivating component. When Ba2+ was used as the charge carrier, the deactivating component should have been decreased.
- In Figure 2B, no labeling of vertical axis (e.g., current density) was found.
- Please appropriately comment on and cite this paper (PMID: 19855067). Also, make a comparison. That paper notably presented the theoretical results for the G406R CaV1 mutant, which are relevant to the present findings.
Minor comments:
- In line 11, please change UK to “United Kingdom” for consistency.
- What is the correct concentration of intracellular Ca2+, mM or mM (e.g., in lines 394 and 396) used throughout the text in the manuscript? Also, is it 0.001 mM (line 156) or 1 mM (lines 149-150)? Please correct or standardize the units used.
- In line 154, what does “mock” mean?
- In line 165, what was “NMDA” prepared? What is the osmolarity in this pipette solution?
- In line 302, “1Hz” needs to be replaced with “1 Hz”.
- In line 175, can Ca2+ current inactivation be appropriately measured as the time constants (t) of current inactivation, because of its time-dependent property in response to square-pulse depolarization?
- In line 331, “Ca2+ and Ba2+ currents” was appropriately used and described. Two bar graphs (Ca2+ and Ba2+ currents) should be presented.

non-available
Round 2
Reviewer 1 Report
Comments and Suggestions for Authors
I would like to thank all authors for addressing my comments and revising their manuscript. I have no further comments.
Author Response
Thank you for your useful and constructive comments.
Reviewer 2 Report
Comments and Suggestions for Authors
In Figures 2B, please show the trail current as the potential was returned to -70 mV. In Figure 3G, the current trace for Ba2+ current could be inappropriate, particularly at the tail deactivating component. When Ba2+ was used as the charge carrier, the deactivating component should have been decreased.
Comments on the Quality of English Language
non-available
